# Spectral Photon-Counting CT Technology in Chest Imaging

**DOI:** 10.3390/jcm10245757

**Published:** 2021-12-09

**Authors:** Salim Aymeric Si-Mohamed, Jade Miailhes, Pierre-Antoine Rodesch, Sara Boccalini, Hugo Lacombe, Valérie Leitman, Vincent Cottin, Loic Boussel, Philippe Douek

**Affiliations:** 1INSA-Lyon, University of Lyon, University Claude-Bernard Lyon 1, UJM-Saint-Étienne, CNRS, Inserm, CREATIS UMR 5220, U1206, 69621 Lyon, France; rodesch@creatis.insa-lyon.fr (P.-A.R.); sara.boccalini@chu-lyon.fr (S.B.); Hugo.Lacombe1@phelma.grenoble-inp.fr (H.L.); vleitman77@gmail.com (V.L.); loic.boussel@chu-lyon.fr (L.B.); philippe.douek@chu-lyon.fr (P.D.); 2Radiology Department, Hospices Civils de Lyon, 69500 Lyon, France; jade.miailhes@gmail.com; 3National Reference Center for Rare Pulmonary Diseases, Louis Pradel Hospital, Hospices Civils de Lyon, UMR 754, INRAE, Claude Bernard University Lyon 1, Member of ERN-LUNG, 69677 Lyon, France; vincent.cottin@chu-lyon.fr

**Keywords:** thorax, lung, diagnostic imaging, computed tomography, photon-counting detectors

## Abstract

The X-ray imaging field is currently undergoing a period of rapid technological innovation in diagnostic imaging equipment. An important recent development is the advent of new X-ray detectors, i.e., photon-counting detectors (PCD), which have been introduced in recent clinical prototype systems, called PCD computed tomography (PCD-CT) or photon-counting CT (PCCT) or spectral photon-counting CT (SPCCT) systems. PCD allows a pixel up to 200 microns pixels at iso-center, which is much smaller than that can be obtained with conventional energy integrating detectors (EID). PCDs have also a higher dose efficiency than EID mainly because of electronic noise suppression. In addition, the energy-resolving capabilities of these detectors allow generating spectral basis imaging, such as the mono-energetic images or the water/iodine material images as well as the K-edge imaging of a contrast agent based on atoms of high atomic number. In recent years, studies have therefore been conducted to determine the potential of PCD-CT as an alternative to conventional CT for chest imaging.

## 1. Introduction

Chest imaging is constantly evolving, and the demand for diagnostic performance is growing in many areas, such as lung cancer, detection of pulmonary nodules, chronic obstructive pulmonary disease, as well as evaluation and follow-up of interstitial lung diseases (ILD). While CT is the most powerful tool currently available for this purpose, it has many limitations such that accurate diagnosis often requires additional histological analyses or other non-radiological examinations.

Photon-counting detector computed tomography (PCD-CT) technology is a new CT modality that has gained increasing interest in all areas of imaging, including musculoskeletal, digestive, cardiovascular, pulmonary, and molecular imaging [1,2,3,4,5,6]. Contrary to energy integrated detectors (EID) found in current standard CT scanners, PCD are made of compounds, such as silicon, cadmium-telluride, or cadmium-zinc-telluride, to directly convert each X-ray photon into an electric pulse, allowing resolution of the energy of the photons. This characteristic allows better intrinsic technical performance by reducing noise, increasing spatial resolution, decreasing beam hardening, and decreasing X-ray doses [7,8]. In addition, the energy-resolving capabilities of these detectors allow to generate spectral basis images, such as the monoenergetic images or water/iodine material images as well as the K-edge images of a contrast agent based on atoms of high atomic number. Taken together, PCD-CT provides new diagnostic perspectives compared to the conventional EID found in current CT systems. Studies have therefore been conducted in recent years, mainly limited to non-contrast lung imaging, to determine the potential of PCD-CT as an alternative to conventional CT. In the present review, we will discuss these studies and explore the perspectives of PCD-CT for chest imaging.

## 2. Technical Aspects

### 2.1. Photon-Counting Detector Technology

PCD are a recently developed technology seen as the future of X-ray measurement; the design of these detectors differs from that of EID, which are commonly used in clinical practice (Figure 1). The EID use a scintillator material transforming the incoming X-ray into visible light that is then captured by a photo-diode. The deposited energy in the pixel is integrated by an application-specific integrated circuit (ASIC) that produces a current proportional to the energy of the incoming photon. In contrast, the PCD are made of a semi-conductor material that directly converts the incoming photon to electrical charges, which migrate into a counting ASIC (Figure 1). This process is called direct conversion (by opposition to indirect conversion). The ASIC shapes a voltage pulse proportional to the incoming photon energy. From the amplitude of the pulse the PCD can differentiate photons according to their energy (Figure 2). PCD-CT can operate in two different modes: conventional or spectral. In the conventional mode, photons are not discriminated according to their energy but are only summed; “conventional images” are those computed from the attenuation of all counted photons. This corresponds to the single energy CT (SECT). The direct conversion design fundamentally improves three major aspects of X-ray SECT imaging: the size of the detector pixel, the reduction of the electronic noise, and the energy weighting in the analogical output signal. The resulting benefits of these aspects are detailed in the next section.

### 2.2. Benefits for Conventional Images

#### 2.2.1. Spatial Resolution

Because the scintillator can produce visible photons in each direction, the EID pixels require separation by a reflector material (Figure 1). This leads to an interseptal gap between pixels that critically decreases the ratio of effective sensitive area to detector area when the pixel size becomes too small. The subsequent major loss of efficiency with respect to exposure limits the size of EID to 0.5 mm at iso-center in normal resolution for most of the current clinical system. PCD pixel pitch is not technically limited and can reach 0.19–0.225 mm at the iso-center in the latest full-body prototypes [9,10,11,12].

The most recent ultra-high-resolution (UHR) EID-CT uses EID made by a different manufacturing process than conventional ones [13]; the septa gap is thinner and closer to the thickness of the anti-scatter collimators (Figure 1). This leads to 0.25-mm voxel element at iso-center, which fills the gap between standard EID and PCD in terms of reconstructed voxel size for lung imaging [14]. However, some methods have shown promising results to correct scatter without a pre-detector collimator [15], even for PCD [16]. This anti-scatter grid removal would increase the sensitive area of PCD and not EID. Moreover, interseptal gaps prohibit the possibility for subpixel resolution, which is a process that better estimates the photon reaching point for a pixel based on the adjacent pixels [17].

To the best of our knowledge, a comparison of the dose efficiency between EID and PCD for a large detector panel with 0.25-mm voxel element size is not yet available. Even with the same this geometry configuration, the PCD still benefit from technical advantages described in the next subsections.

#### 2.2.2. Reduction of Electronic Noise

According to the number of available thresholds (between 3 and 8 according to PCD manufacturer), several measurement bins can be defined (Figure 2) [9,11,12]. The first threshold is set just above the electronic noise level and suppresses it from the final counts, which cannot be done for EIDs, and therefore, electronic noise is added to the integrated energy. This means the noise in the reconstructed image is lower with PCD than with EID for the same dose.

#### 2.2.3. Contrast Improvement

In addition to noise reduction, another aspect improving contrast-to-noise ratio (CNR) is the energy weighting of each incoming photon. As the linear attenuation coefficient decreases with energy, lower energy provides greater contrast, for example, when differentiating water from calcium [18] (Figure 3). Whereas EID produce a signal proportional to the incoming energy for one photon (linear weight), PCD will generate only one count irrespective of the photon measured (constant weight).

#### 2.2.4. Reduction of Beam Hardening

The constant weight also reduces beam hardening artifacts in conventional images. Beam hardening is due to a difference in mean energies between the incident and the attenuated spectra, and the proportional weighting of EID increases this difference, which is not the case for the constant weighting of PCD [8,19].

#### 2.2.5. Dose Efficiency

The advantages described above of PCD over EID lead to a better dose efficiency, as demonstrated by several studies [9,20,21,22,23]. The effect of this dose efficiency can be either to produce the same image quality for the same dose or to investigate new protocols. In addition, iterative reconstruction can decrease noise levels in reconstructed CT images. Combined with the emergence of iterative reconstruction, PCD-CT enables either the reconstruction with a larger matrix (1024 × 1024 or 2048 × 2048) or the efficiency of ultra-low dose protocols [9,22].

### 2.3. Spectral Mode

Recently, progress in informatics (high storage capacity, high-speed computation reconstruction) has allowed for spectral reconstructions in clinical practice through dual-energy CT (DECT) devices. However, DECT systems require major modification in their architecture compared to single energy CT [1]. While PCD-CT has the advantage to be able to run in both conventional and spectral mode, using a traditional X-ray source and CT geometry allows different types of images that are described in following subsections.

#### 2.3.1. Virtual Monoenergetic (VMI) Images

PCD have the possibility to provide spectral measurement bins by differentiating incoming photons according to their energy (Figure 2). This spectral information can be used to reconstruct VMI [23]. VMIs are computed from spectral information at a (virtual) selected energy leading to beam hardening artifact reduction. Conversely, conventional images are the reconstruction of the attenuation coefficient averaged over the measured spectrum. VMIs can be reconstructed from either a PCD or a DECT acquisition. At low energies, VMIs allow a boost of the photoelectric effect such as the iodine boost in the presence of an iodinated contrast agent. This may allow a better depiction of the vessel lumen [24], while at high energies, VMI are robust to metal artifacts [25]. Another metal artifact-reduction technique is the reconstruction of an image using only photons measured over 70 keV. The resulting high-threshold image presents an increased CNR compared to conventional images in presence of metal artifacts [26].

#### 2.3.2. K-edge Imaging

K-edge materials are a type of material with a discontinuity in their attenuation coefficient (Figure 4). This spectral singularity allows for an accurate contrast agent identification in the reconstructed volume [27]. Imaging of a K-edge material, known as K-edge imaging, requires the measurement of at least three energy bins, which is not feasible with DECT [1]. To be detected for clinical imaging, this effect must be within the energy range of the diagnostic X-ray that is between 40 and 140 keV. Unfortunately, iodine is not a good candidate because of the photon starvation around its K-edge, which is 33.3 keV. Hence, new contrast agents for CT should be designed and validated for K-edge imaging using high atomic-number atoms, such as gadolinium, gold, or bismuth (Figure 4).

### 2.4. Conclusions

In this section detailing technical aspects, we introduced the three main advantages of PCD over EID for conventional imaging: smaller detector pixel, electronic noise reduction, and more efficient energy weighting of photons. These advantages have been demonstrated to improve conventional image quality in pre-clinical and preliminary clinical studies. However, this is not the only strong point of PCD-CT, as it can perform simultaneously spectral mode to study VMIs and K-edge contrast agents, which may open to new and promising applications.

## 3. Pre-Clinical and Clinical Applications

### 3.1. Parenchyma Imaging

Despite the recent availability of high-resolution CT (HRCT), evaluation of lung parenchyma integrity is a daily challenge for the radiologist. The principal reason is that radiological evaluation is sometimes limited by the fine semiology of the anatomical structures of the lung lobule [28]; pathological analysis therefore remains the technique of choice in case of diagnostic uncertainty [29]. However, due to its high spatial resolution, PCD-CT systems could revolutionize the evaluation of the lung. Accordingly, numerous studies have investigated PCD-CT imaging of the parenchyma by studying the influence of the reconstruction parameters, such as the matrix size, the slice thickness, and the iterative algorithms recently made available for this new technology.

For instance, Bartlett et al. [30] demonstrated in patients the value of using a 1024 matrix with a high-frequency filter (Q65) with a limited field-of-view (FOV) PCD-CT system, in comparison with a 512 and 1024 matrix and standard detailed filter (B46) with PCD-CT and EID-CT. These reconstruction parameters were used to convey the high spatial resolution of the PCD-CT system, which improved the depiction of the lung parenchyma structures, such as the bronchi and morphological features of lesions (e.g., nodule, ground-glass nodules). Recently, on a different PCD-CT system [9], we demonstrated in human volunteer with a large FOV and a 1024 matrix, that lung structures, such as fissures, distal airways, and vessels, were of greater conspicuity and sharpness with better overall image quality than standard CT, despite a 23% flux reduction (Figure 5). Interestingly, the noise was also better rated by three experienced chest radiologists who noted a particular texture that characterizes high-frequency noise. In this study, we showed that the UHR parameters allowed to depict a 178-µm line width on a line pair phantom, which was enabled by a spatial resolution at 22.3 lp/cm at 10% modulation transfer function. Similarly, Leng et al. [10] reported in patients with a different PCD-CT system the value of an UHR mode for small lung structures using a sharp filter with a spatial frequency cutoff of 32.4 lp/cm, a 1024 matrix, and 0.25 slice thickness in comparison to a standard (macro) and sharp mode. The sharp and UHR modes showed similar modulation transfer functions, both better than for macro mode; 10% modulation transfer function value was 9.48 lp/cm for macro, 16.05 lp/cm for sharp, and 17.69 lp/cm for UHR modes.

Taken together, these studies show the potential for more accurate parenchyma evaluation in vitro as well as in human volunteers and patients with the UHR mode of the PCD-CT technology that requires confirmation in a prospective clinical study.

### 3.2. Nodule Imaging

Lung nodule imaging is a challenging task that necessitates an accurate characterization of the morphological features of a nodule [31]. In recent years, several studies have proven the performance of PCD-CT in the detection and morphological characterization of pulmonary nodules, thus demonstrating its potential for this public health problem. For example, Zhou et al. [32] studied the impact of UHR mode for lung nodule volume evaluation and shape characterization using a PCD-CT system. They used a 512 matrix with a small FOV (11 cm) resulting in a pixel size of 215 µm and compared the UHR mode to the macro mode using two reconstruction algorithms (s80f and b46f). The UHR mode used an effective detector pixel size of 0.25 mm by 0.25 mm at the iso-center, while the conventional macro mode was limited to 0.5 mm by 0.5 mm. For small nodules (diameter ≤ 5 mm), UHR mode was able to achieve more accurate volume measurements than macro mode due to higher spatial resolution, which is an advantage for routine use. Moreover, all acquisitions showed comparatively lower bias in the evaluation of the volume of spherical nodules. Additionally, for irregularly shaped (star) nodules, particularly of small size (diameter ≤ 5 mm), UHR acquisitions with the sharp kernel provided substantially better accuracy in volume measurements compared to the other three acquisition mode/reconstruction kernel combinations. Furthermore, receiver-operating-characteristics analyses showed a clear benefit of the UHR mode, demonstrating an increased ability to differentiate small, smooth, spherical nodules from small, irregular-shaped nodules. These finding are particularly important, as studies have shown that nodules with irregular or spiculated margins, particularly with distortion of adjacent vessels, are likely to be malignant.

In a second study, Zhou et al. [33] performed a similar study in phantoms with similar parameters and proved that for all nodules, the volume estimation was more accurate, as there was a lower mean absolute percent error (6.5%) compared with macro mode (11.1% to 12.9%). The improvement of volume measurement with the UHR mode was more obvious for small nodule size (3 mm, 5 mm) or star-shaped nodules. Thus, the authors demonstrated that the UHR mode of the evaluated PCD-CT system was able to improve measurement accuracy for nodule volume and nodule shape characterization.

In line with these studies, Kopp et al. [34] demonstrated a better morphological and volumetric evaluation of different nodule phantoms presenting spikes with a different PCD-CT system. The volume estimation indicated better accuracy for PCD-CT compared to CT and HR-CT (with a root mean squared error of 21.3 mm^3^ for SPCCT, 28.5 mm^3^ for CT, and 26.4 mm^3^ for HR-CT). The Dice similarity coefficient was greater for PCD-CT considering all nodule shapes and sizes (mean, PCD-CT 0.90; CT: 0.85; HR-CT: 0.85). These findings were explained by the higher spatial resolution of the system as demonstrated by a higher modulation transfer function (10% modulation transfer function at 21.7 lp/cm).

Recently, using a clinical prototype [22], we demonstrated the enhanced performances of PCD-CT for solid and ground-glass lung nodule detectability in a phantom study using a task-based model observer assessment and a subjective image quality evaluation. The comparison between a dual-layer CT and the PCD-CT system from the same manufacturer with optimized parameters for each platform found a noise reduction with PCD-CT, as demonstrated by the noise power spectrum analysis and an improved spatial resolution as demonstrated by the task transfer function. As a consequence, the task-based model observer (*d’*) evaluation demonstrated a greater detectability for solid and ground-glass nodules (Figure 6). It is noteworthy that the difference in detectability between CT systems was more pronounced for the ground-glass nodules, which are known to be more difficult to detect due to their low contrast with conventional CT systems. Finally, experienced chest radiologists determined that the best compromise for optimal image quality, noise reduction, and increased spatial resolution was reached by using the iterative reconstruction at level iDose 6, which may enable a clinical use.

Finally, Jungblut et al. [35] evaluated the image quality of pulmonary nodule at different low radiation dose for PCD-CT and EID-CT systems. Their analysis was based on a subjective evaluation by observers and on a commercial artificial intelligence-based, computer-aided detection system. They analyzed an anthropomorphic chest phantom containing 14 pulmonary nodules of different sizes with three vendor-specific scanning modes at decreasing matched radiation dose levels. By comparing PCD-CT to EID-CT images, they found using the HR mode a better image quality with PCD-CT in the subjective analysis as well as the lowest image noise in favor of a better objective analysis. Despite the improved performance, the artificial intelligence-based, computer-aided detection system delivered comparable results for lung nodule detection and volumetry between PCD- and dose-matched EID-CT. Nevertheless, the mean sensitivity PCD-CT images was higher than with EID-CT with, for example, a rate of 95% versus 86% for dose-matched EID-CT with a volume CT dose index of 0.41 mGy.

Taken together, these studies show the potential for more accurate lung nodule evaluation in vitro and in vivo with PCD-CT technology that requires confirmation in a prospective clinical study.

### 3.3. Lung Cancer Screening Imaging

Lung cancer is the leading cause of cancer deaths worldwide [36], and despite the conclusions of the National Lung Screening Trial that CT screening significantly reduces mortality associated with lung cancer, among other diseases [37], to date, there is no worldwide agreement on the screening recommendations. This can in part be explained by the fact that the diagnostic benefits of lung cancer screening have to be balanced with the inherent risks of ionizing radiation as well as overdiagnosis and work-up of false-positive findings [38].

Symons et al. [39] studied the impact of a low-dose acquisition with PCD-CT and a conventional CT with a 512 matrix on two phantoms by comparing attenuation accuracy, noise, and CNR values between different dose settings. Hounsfield unit stability and accuracy for lung, ground-glass nodules, and emphysema were greater with PCD-CT than EID-CT, particularly at lower doses where the attenuation decreased significantly with EID-CT. In addition, a reduction of noise was consistently found for PCD-CT at 80, 100, and 120 kVp in comparison to EID-CT. As a result, CNR were improved compared to EID for ground-glass nodules and emphysema, particularly for 100 and 80 kVp, without significant difference for other objects (acrylic, water, lung foam, air, etc.). These results support the technical benefits of PCD making PCD-CT a promising tool in radiation dose optimization that is critical in further improving the risk-benefit ratio of CT lung cancer screening; however, they require confirmation in a prospective clinical study.

### 3.4. Low and Ultra-Low Dose Imaging

Ultra-low and low-dose CT has been increasingly used in low-radiation risk assessment of lung cancer, but this is not the only pulmonary disease where the risk-benefit ratio seems positive; for instance, the follow-up of fibrosing disease at low doses of radiation could be of great interest because of the iterative CT scans. However, the main limitation for use of these protocols is the impairment of the objective and subjective image quality because of the photon starvation and increase in artefacts, such as beam hardening using EID-CT. With PCD-CT, numerous studies have suggested an improvement of the image quality (Figure 7 and Figure 8). For instance, Symons et al. [39] proved on phantoms using a 512 matrix the capacity of PCD technology to be more stable in measuring the parenchyma attenuation at low dose. This may also provide better reproducibility and accuracy for quantitative imaging in the future, such as performed for lung density. The latter is necessary to monitor disease, such as alpha-1 antitrypsin deficiency, emphysema, and idiopathic pulmonary fibrosis, but can also be used for the treatment strategy, such as performed in alpha-1 antitrypsin deficiency [40]. More recently, the same group reported in 30 healthy volunteers the feasibility of a CT lung cancer screening protocol (120 kVp, 20 mAs) with a 512 matrix and an iterative reconstruction algorithm [41]. They showed that experienced readers identified the PCD-CT images as having better diagnostic quality for assessment of lung tissue, lung nodules, soft tissue, and bone with lower subjective image noise when compared to those obtained with the EID system. They also found a better image quality in areas with known beam hardening, e.g., around the vertebrae and in the apical lobes. Quantitative measurements showed that PCD-CT images had 15.2–16.8% lower noise at two different dose levels, with 21.0% higher lung nodule CNR. In addition, we demonstrated the feasibility of using low dose (CTDI: 1.11 mGy) and ultra-low dose (CTDI: 0.10 mGy) protocols in lung nodule evaluation with phantoms and one human volunteer while preserving UHR parameters [9]. We used a 1024 matrix, a large FOV, and slice thickness of 0.25 mm and found a satisfactory detection with low X-ray dose protocol for solid, mixed, and ground-glass nodules. Using ultra-low dose parameters (80 kVp, 3 mAs), ground-glass nodules were not clearly defined because of increasing noise. However, iterative reconstruction (i.e., iDose 5 and 9) enabled a satisfactory nodule visualization. Moreover, improved image quality was found with low-dose PCD-CT images compared to standard-dose EID-CT for lung imaging in a human volunteer despite 87% of flux reduction. Additionally, the use of newly developed iterative reconstruction for PCD-CT data led to a decrease in noise by up to 68% when using iDose reconstruction algorithm (iDose 9). In conclusion, this study showed the feasibility of significant dose reduction with PCD-CT with or without iterative reconstruction while conserving the high-resolution parameters for lung analysis.

Taken together, these studies highlight the feasibility of PCD-CT imaging for improving lung nodule detection with a good compromise between increased spatial resolution, decreased dose, and equivalent or reduced noise.

### 3.5. Interstitial Lung Disease Imaging

Interstitial lung diseases are complex pathologies and HR-CT evaluation is essential for accurate diagnosis and determination of the best treatment. However, this requires very high spatial resolution because of the microsemiology of their key signs, e.g., intralobular reticulations, bronchiectasis, and honeycombing, which are frequently indeterminate and may lead to invasive procedures, such as biopsies for histological evaluation [42]. In addition, current low-dose protocols, i.e., lower than 1 mSv, are not recommended due to the impaired image quality that may lead to misclassification of the interstitial lung disease [29]. Therefore, PCD, because of its high resolution and noise reduction capacity, could be a promising tool for ILD evaluation.

In an attempt to evaluate the feasibility of PCD-CT in interstitial lung diseases, a recent study conducted by Ferda et al. [12], with a clinical prototype large FOV PCD-CT system, identified a few cases in a cohort of 60 patients addressed for various reasons. They investigated the diagnostic quality of PCD-CT lung imaging using a standard and a low-dose protocol by subjectively assessing critical structures, such as bronchi and bronchial walls. When PCD-CT images were compared to those obtained with a similar dose using EID-CT, the background noise was significantly lower, and the signal-to-noise ratios were significantly higher in PCD-CT images. An important point is that the subjective quality score between full and low-dose lung imaging was comparable. In patients with interstitial lung disease, the PCD-CT demonstrated a better visualization of higher-order bronchi and third-/fourth-/fifth-order bronchial walls, such as previously suggested with a PCD-CT system from a different manufacturer [9]. However, the authors did not evaluate the diagnostic performance of PCD-CT compared to EID-CT for key signs of fibrosis. Nevertheless, these initial findings indicated an important potential for further radiation dose reduction in interstitial lung disease imaging diagnosis and follow-up.

### 3.6. Distal Airways and Bronchial Imaging

Bronchial diseases are difficult to evaluate with conventional CT imaging. For patients with chronic obstructive pulmonary disease, the initial morphological evaluation of the bronchial tubes is difficult as well as their follow-up under treatment due to the lack of spatial resolution of conventional scanners. This limitation is also observed for interstitial lung disease, where the differentiation between traction bronchiectasis and honeycombing leads to numerous controversies in the literature [42], leaving a room for image quality improvement using PCD-CT.

For example, Bartlett et al. [30] included 22 patients with pulmonary condition (pneumonia, pulmonary nodule, etc.) and showed improvement in the detection of higher-order bronchi compared with the EID-CT system, irrespective of the image reconstruction kernel used (readers detected more seventh- and eighth-order bronchi). The addition of a sharp reconstruction kernel designed to convey the higher spatial resolution of the PCD-CT system (Q65) further improved the visualization of small bronchi and bronchial walls (walls of the third- and fourth-order bronchi in both lungs). The results of this study demonstrated the potential benefit of PCD-CT lung HR-CT, particularly in assessing airway diseases. Similarly, Kopp et al. [34] compared in-vivo images of a rabbit and a patient acquired with PCD-CT and HR-CT. With HR-CT, bronchi and bronchioles down to a diameter of 1.5–2 mm could be identified. Comparing images of HR-CT and PCD-CT adjusted to the same size, vessels, and walls of bronchioles could be visualized more distinctly with PCD-CT. In addition, we showed a better conspicuity and sharpness of the distal airways as rated by three experienced chest radiologists on a human volunteer, which allowed to see the origin of a 0.5-mm diameter bronchiole that was not visible on an EID-CT [9]. Taken together, these studies support the potential of PCD-CT to propose a new evaluation of bronchial diseases (Figure 9).

### 3.7. Pulmonary Vascularization Imaging

Pulmonary vascular involvement is highly prevalent in various lung diseases, such as in chronic obstructive pulmonary disease, emphysema, chronic thromboembolic pulmonary hypertension, as well as infectious disease, such as COVID-19 [43,44,45,46,47]. Current standard CT can allow the diagnosis of vessel structure abnormalities, such as thrombus or caliper disparity, while DECT enables the specific imaging of the iodine distribution in the lung, allowing the microcirculation imaging known as a surrogate marker of lung perfusion [48]. Considering the probable improved performance of PCD-CT compared to EID-CT, there is hope for better monitoring the proximal and distal pulmonary vascular involvement in the diseases cited above (Figure 10). For example, this would be of great interest in chronic thromboembolic pulmonary hypertension for the characterization of the involvement of microvasculopathy in the sub-pleural area that might contribute to severe hemodynamics, as suggested by Onishi et al. [49]. In addition, the virtual monoenergetic capabilities of PCD-CT would allow the use of low monoenergetic images, e.g., 40 or 50 keV, to boost iodine attenuation that would help decrease the amount of iodinated contrast agent for a pulmonary CT angiography.

### 3.8. K-Edge Imaging

Through its multi-energy CT capabilities, PCD technology enables a spectral tissue characterization and differentiation with or without the use of contrast agents, explaining the use of the term spectral photon-counting CT (SPCCT) in recent studies [1,50,51,52,53,54,55,56,57,58,59,60,61,62,63,64,65,66]. Among these capabilities, K-edge imaging is one of the new features of SPCCT enabled by the energy-resolving characteristics of PCD. It is based on the recognition of the specific K-edge signature of an atom, i.e., the binding energy between the inner electron shell and the atom [67]. Indeed, by dividing the spectrum into well-chosen energy-based datasets, it is possible to detect multiple elements, such as gadolinium, gold, and other atoms that have a K-edge within the relevant energy range of the X-ray spectrum used (e.g., ~40–100 keV; Figure 11) [27]. The main advantage of this technique is to permit a specific and quantitative imaging of the agent without any signal arising from the background, which would allow an improved contrast of the tissue as compared to conventional CT imaging. This type of imaging requires the use of contrast agents adapted for SPCCT, which opens the door for research and development. Among the candidate contrast agents for K-edge imaging, nanoparticles are becoming the agents of choice because of their numerous advantages, such as high payload of atoms, potential for blood pool effect, tunable physicochemical properties, potential for targeting imaging, cell tracking, and theranostic applications [50,68,69].

In 2018 [51], we showed in vitro the feasibility of monocolor and bicolor imaging in order to detect and quantify different mixed contrast agents. Three contrast agents (iodine-, gadolinium-, and gold-based) distributed in tubes at varying proportions were tested by reconstructing the specific images of each material, i.e., based on a two-basis material decomposition (MD) for iodine, three-basis MD for gadolinium or gold with or without iodine, and four-basis MD for gadolinium and gold. The mixtures were prepared such that the solutions could not be differentiated in conventional images. However, distinction was observed in the material images within the same samples, and the measured and prepared concentrations were strongly correlated confirming that SPCCT enables multicolor quantitative imaging in vitro.

Additionally, in 2017 [56], we used a first of its kind system with high-count rate capabilities to demonstrate in vivo in rabbits the feasibility of a dynamic monocolor K-edge imaging of a blood pool agent using PEGylated gold nanoparticles. In the organs analyzed, we showed the persistent visualization of lung vascularization during the first hours after injection. In addition, we presented for the first time the feasibility of bicolor K-edge imaging in animals, i.e., simultaneous differentiation of two different contrast agents, using gold nanoparticles and an iodinated contrast agent (Figure 12). Quantification of these contrast agents in the lungs demonstrated persistent concentration for gold, while for iodine, an arterial first-pass was noticed with rapid clearance in the lungs, as expected. Taken together, these features may potentially allow a new form of functional imaging, where multiple contrast agents with different pharmacokinetics are used simultaneously in the same biological system.

## 4. Limitations and Perspectives

The studies cited above provide insight for the future potential of PCD-CT imaging, particularly in-vivo studies that confirm the improvement in spatial resolution and noise management. However, they remain preliminary, and larger patient cohort studies are still needed to identify all the clinical benefits of this new technology. Despite this, it is already possible to consider the clinical potential of SPCCT thanks to studies such as that reported by Yanagawa et al. [70], who proved the value of studying lung adenocarcinomas with a high spatial resolution (2048 matrix on a conventional CT) to define local invasion of the tumor before the surgery.

With respect to the K-edge imaging feature of SPCCT, there are still many hurdles to overcome before considering a clinical translation. These are related to the detection chain and spectral model performance but also by the current contrast agents. It should be noted that, for example, the gadolinium doses needed for in-vivo imaging are far higher than currently used and recommended in MR, and other contrast agents are in an experimental stage that do not allow any kind of clinical human use in the nearer future.

In addition, the current technical limitations of the PCD-CT systems that depend on the progress made by manufacturers should be noted, such as the detector array number, the rotation time, and the energy-resolving capabilities. However, these limitations are expected to be addressed in the near future.

## 5. Conclusions

Photon-counting detector computed tomography (PCD-CT) represents an emerging medical imaging modality. The key features are an improved spatial and contrast resolution and a significant noise reduction in comparison to the standard energy-integrating detectors CT (EID-CT) as well as spectral capabilities, such as K-edge imaging, that may contribute to the improvement of current CT practice in chest imaging.

## Figures and Tables

**Figure 1 jcm-10-05757-f001:**
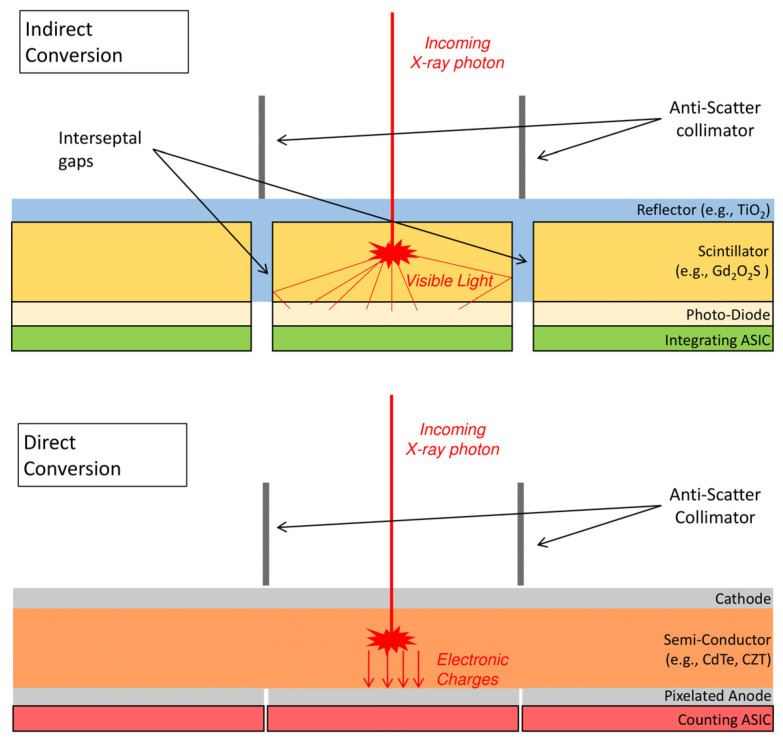
Schematic representation of photon detection technologies: energy-integrating (**top**) and photon-counting detectors (**bottom**).

**Figure 2 jcm-10-05757-f002:**
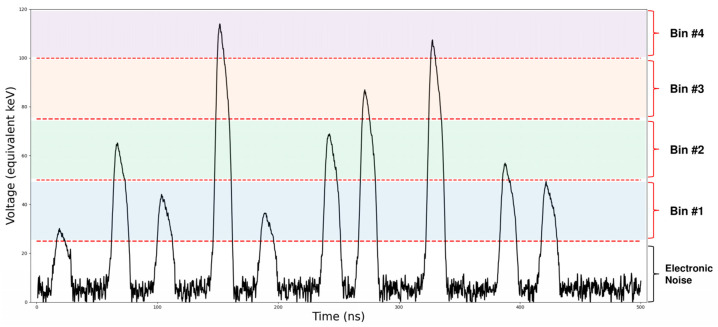
Example of a 500 ns signal output of a photon-counting detector pixel.

**Figure 3 jcm-10-05757-f003:**
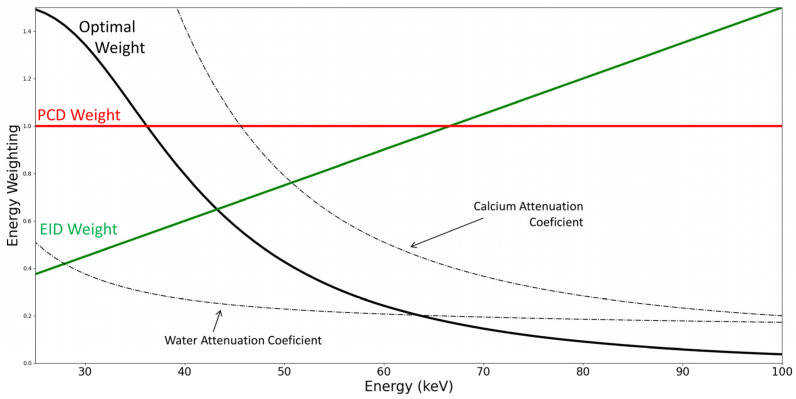
Energy weighting of energy integrating detectors (EID) and photon-counting detectors (PCD). The optimal weight for differentiating calcium from water is derived from the attenuation coefficients subtraction (dashed lines).

**Figure 4 jcm-10-05757-f004:**
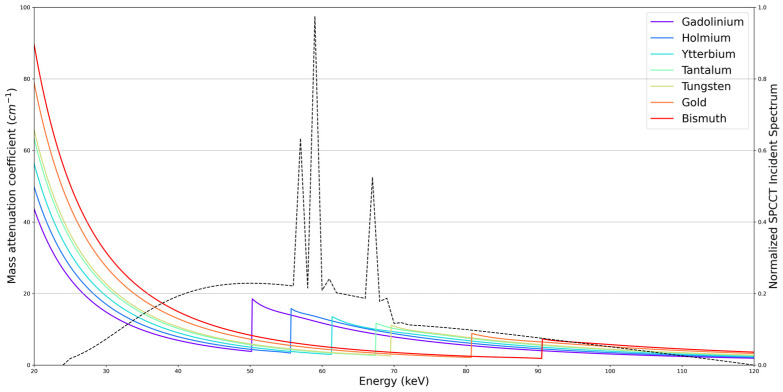
Attenuation coefficients of contrast agents with a K-edge in the medical energy range.

**Figure 5 jcm-10-05757-f005:**
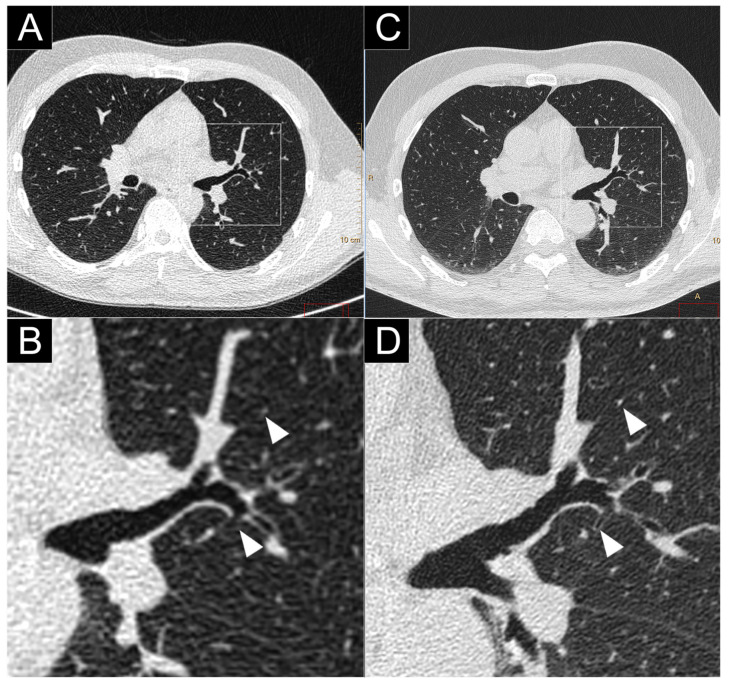
Comparison of high-resolution lung imaging between a conventional CT (Brilliance 64; Philips Haifa, Israel; (**A**,**B**)), a clinical prototype PCD-CT (SPCCT; Philips; (**C**,**D**)), and in a human volunteer. Close-up views on the left pulmonary hilum found a greater overall image quality as well as better depiction and a greater number of small structures (bronchial wall, vessels; white arrows).

**Figure 6 jcm-10-05757-f006:**
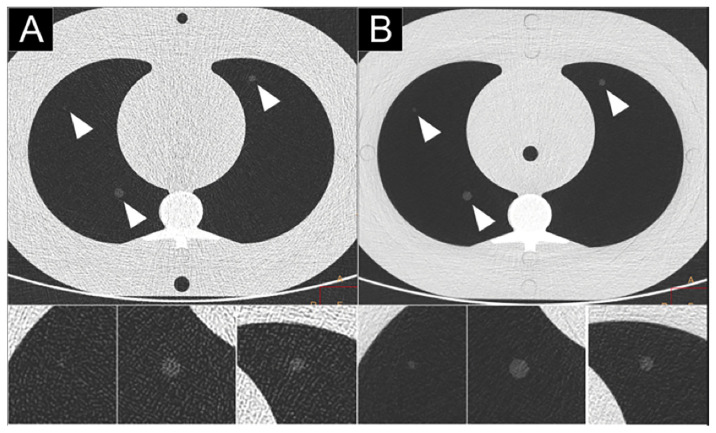
Comparison of high-resolution CT lung imaging of ground-glass nodules (2, 4, 6 mm) between a dual-layer EID-CT (iQon; Philips, Haifa, Israel) (**A**) and a clinical prototype PCD-CT (SPCCT; Philips) (**B**) in an anthropomorphic thorax phantom with an extension ring simulating an obese patient, using a standard dose protocol (120 kVp, 40 mAs).

**Figure 7 jcm-10-05757-f007:**
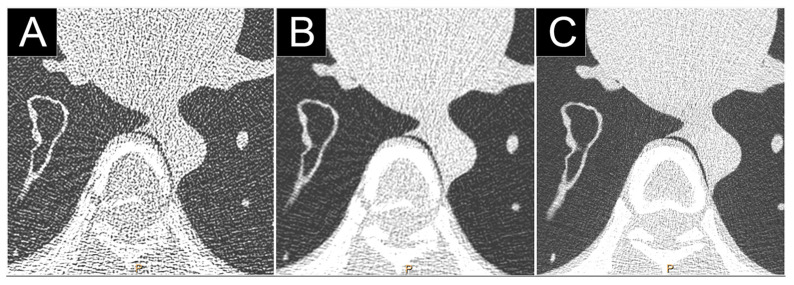
Comparison of high-resolution low dose lung imaging (120 kVp, 10 mAs) between a dual-layer EID-CT (iQon, Philips, Haifa, Israel; (**A**): filter YB, (**B**): filter F) and a clinical prototype PCD-CT (Philips; (**C**): filter detailed 1) in an anthropomorphic phantom CT Torso CTU-41 (Kyoto Kagaku, Tokyo, Japan). Both 1024 matrix and field-of-view of 350 mm were matched. Beam hardening was greatly reduced on the PCD-CT images, greatly improving the image quality.

**Figure 8 jcm-10-05757-f008:**
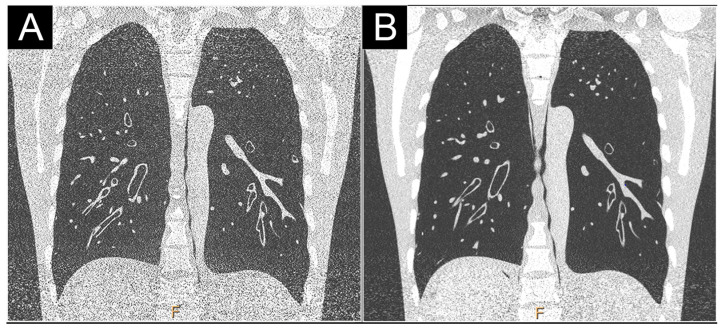
Comparison of high-resolution low dose lung imaging (120 kVp, 10 mAs) between a conventional dual-layer CT (iQon; Philips, Haifa, Israel; (**A**): filter YB) and a clinical PCD-CT (Philips; (**B**): filter detailed 1) in an anthropomorphic phantom CT Torso CTU-41 (Kyoto Kagaku, Tokyo, Japan). Both 1024 matrix and field-of-view of 350 mm were matched. Noise was significantly reduced on the PCD-CT images in the upper and basal lobes, greatly improving the image quality.

**Figure 9 jcm-10-05757-f009:**
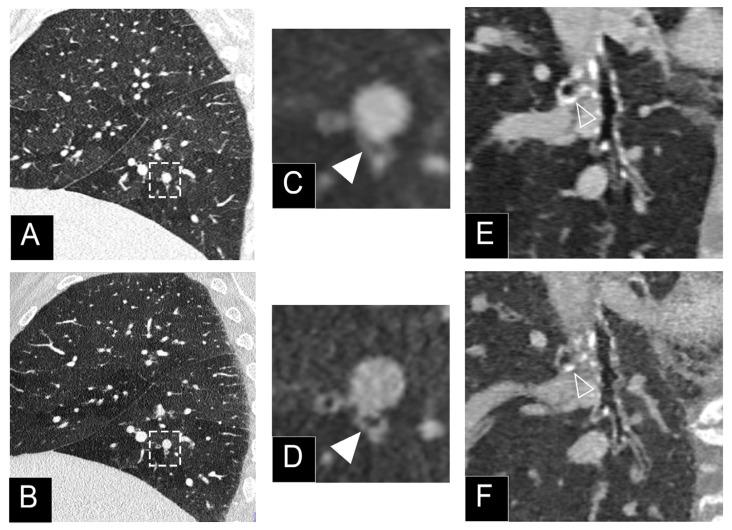
Comparison of the airways imaging in a 72-year-old patient with a respiratory bronchiolitis, which was on the same day as a clinical prototype PCD-CT (first row; Philips, Haifa, Israel) and a dual-layer EID-CT (second row; iQon, Philips). Sagittal views (**A**,**B**) demonstrated thickening of the bronchial wall as well as a mosaicism probably due to air trapping. Close-up views of the airspaces (**C**,**D**) showed a better sharpness and conspicuity of the bronchial wall (white arrowheads) with PCD-CT (**D**). Greater sharpness of the bronchial calcifications were noticed on PCD-CT images (**F**) compared to EID-CT images ((**E**); white empty arrowheads).

**Figure 10 jcm-10-05757-f010:**
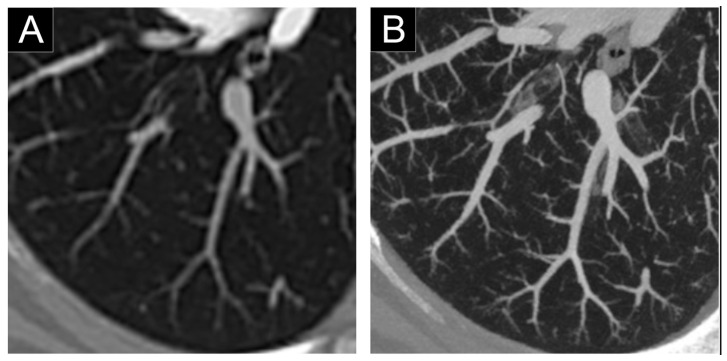
Comparison of the distal arterial pulmonary tree with a dual-layer EID-CT (iQon, Philips, Haifa, lsrael; (**A**)) and a clinical prototype PCD-CT (Philips; (**B**)) and) after injection of iodinated contrast agent. The improvement in quality is visible, notably of the distal vessel lumen and calipers that are depicted until the pleural space.

**Figure 11 jcm-10-05757-f011:**
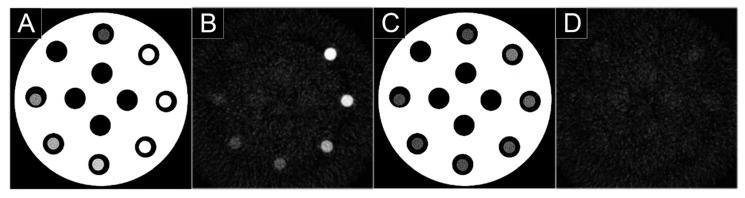
Spectral photon-counting CT (SPCCT) images of a phantom containing tubes with clockwise decrease concentrations of gadolinium from 15 mg/mL to 1 mg/mL of atoms using a clinical prototype (SPCCT; Philips, Haifa, Israel). Conventional image (**A**) and material decomposition of the K-edge image was obtained by reconstructing three material bases (**B**): K-edge image, (**C**): water image, (**D**): iodine image. Only the K-edge images showed the specific signal in the tubes according to the gadolinium concentrations, while water image showed signal from the plastic phantom made of water and the water in the tubes. Iodine image showed no signal accordingly to the absence of iodine.

**Figure 12 jcm-10-05757-f012:**
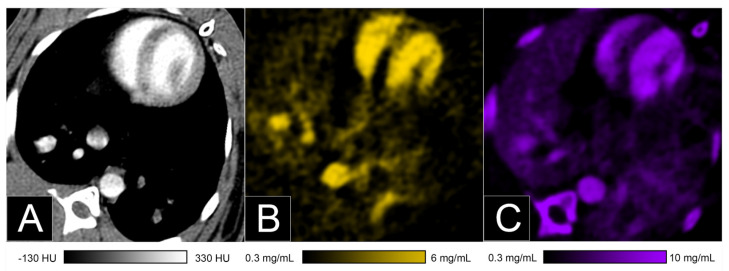
Bicolor imaging of the lung perfusion blood volume in a rabbit after injection of a standard iodinated contrast agent and a contrast agent on a K-edge material (gold nanoparticles) using a spectral photon-counting CT (SPCCT; Philips, Haifa, Israel). Conventional image (**A**) showed the enhancement of the chest vessels as well as the underlying tissue, while the gold K-edge image showed only the signal of gold (**B**). The iodine image showed the signal of iodine as well as misclassification of the bone (**C**).

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
