# Peer review of "Spectral Photon-Counting CT Technology in Chest Imaging"

_jcm, 2021, doi:10.3390/jcm10245757_

Round 1

Reviewer 1 Report

The authors have adressed my concerns. I have no further comments.

Author Response

We thank the reviewer for his comment on the revised manuscript.

Reviewer 2 Report

Attached file contains detailed comments.

Author Response

The authors present a review of the applications of photon counting detectors in the context of thoracic imaging. The review is comprehensive and would be of interest to the readership of Journal of Clinical Medicine. There are, however, some issues which preclude it from publication in its current form.

[1] I am not sure that EIDs have a technical spatial resolution limit of 0.5 mm as a hard rule. I can say from experience that there are higher resolution EIDs in the present day. Line 76-77 should be clarified.

Answer: We agree with the reviewer. Despite the fact that most of the clinical system present a resolution limit of 0.5 mm, recent system with high resolution capabilities overcome this technical limitation with a detector pixel reaching 0.25 mm at isocenter. We thus edited the sentence line 76-77 such as follows: “The subsequent major loss of efficiency with respect to exposure limits the size of EID to 0.5 mm at iso-center in normal resolution for most of the current clinical system”, and added the description of the recent high-resolution EID-CT line 82.

[2] Section 2.2 could have the fact that higher temporal resolution is possible using photon counting detection compared to a standard EID setup.

Answer: We thank the reviewer for his/her comment. Regarding temporal resolution, we are convinced of the absence of difference between PCD and EID respectively to clinical CT system. Temporal resolution depends on the exposure time of each voxel, which in turn depends on rotation time and pitch. The data to be acquired is to collect views from 180 degrees, which requires 200-240 degrees of gantry rotation depending on FOV (need >180 degrees of gantry rotation due to fan to parallel beam conversion).

[3] I think line 94 might be specific to a certain PCD vendor with respect to the number of energy bins possible. Other detectors might have more or less bins possible.

Answer: From far we know, current clinical prototype PCD-CTs from 3 main vendors are presenting from 3 to 8 bins as detailed in the manuscript. However, only one is now available for commercial use.

[4] I think an important consideration with PCDs is the cost relative to a EID currently. It is a limiting factor of creating huge detector arrays. I am not sure that there are the big 1024x1024 or 2048x2048 PCD panels as of yet. Perhaps this could be included in the analysis.

Answer: We thank the reviewer for his/her comment. We are not sure to understand the comment in the light of the PCD-CT systems suitable for clinical imaging. That is why we think that availability of big PCD panels is irrelevant for the current manuscript and the systems discussed in this paper.

Regarding cost – PCD is a new technology and as such is indeed more expensive than EID.

However usually costs are dropping as technology matures and as quantities are rising. With the high focus on PCD technology by all vendors it is very likely that cost will not disable this technology from going into wide clinical use.

[5] Other than cost, are there any disadvantages of PCDs relative to EIDs currently?

Answer: We thank the reviewer for his/her comment. The short answer for the long term is no.

In the short term, there are challenges of detector response and pile up that have to be addressed by combination of hardware and software solutions. But these are under active development by all CT vendors who work on PCD technology.

[6] Has there been any research to PCDs in a flat-panel configuration for use in CBCT? Something that could eventually be mounted onto the on-board imager of a clinical linear accelerator in an Oncology context.

Answer: We are not aware of such work. As noted above this paper is regular CT and not about cone beam / flat panel-based CT.

This manuscript is a resubmission of an earlier submission. The following is a list of the peer review reports and author responses from that submission.

Round 1

Reviewer 1 Report

The authors present a review article entitled “Spectral Photon-Counting CT technology in Chest Imaging”. In this review article, the authors summarize current evidence of Photon Counting Detector CT (PCD-CT) imaging for thoracic applications.

Overall, the article is well written and comprehensive. Technical aspects and current studies (focusing mostly on non-contrast lung imaging) are in general well described. However, I do have several comments that should be addressed before this manuscript could be acceptable for publication.

My main concern is that the authors describe the evidence for initial technical validation studies of a technique that is just starting to be commercially available. In addition, there is an absolute lack of clinical data that supports that any technical improvements of PCD-CT translates into any kind of benefit. Most PCD-CT applications have been limited to rather simple CT applications, e.g. impossibility to acquire in vivo, cardiac gated imaging. I am not sure how this matches the need of a journal that is focused on “clinical medicine”. Otherwise, the well structured article with sub-chapters explains comprehensively the benefits of PCD for lung imaging. However, I certainly miss a limitation / critical part discussing current limitations and future aspects that have to be addressed. I agree with the authors that for example k-edge imaging is a highly promising aspect, however it should be noted that for example needed gadolinium doses for in vivo imaging are way higher than currently used and recommended doses in MR and other contrast agents are in an experimental stage that do not allow any kind of clinical human use in the nearer future. Chest imaging could also include information about mediastinum, tumor, tumor perfusion etc. The article is mainly limited to evidence of non-contrast lung imaging. This should be changed or title and or scope should reflect this.

Details and Comments:

  • Abstract is lacking of any relation to spectral imaging, focusing on increased spatial resolution and improved dose efficiency.
  • Introduction: Use of PCD-CT is currently limited to prototype devices that are just becoming commercially available right now – to speak about a boom is overexaggerated. I would recommend to tone down statements.
  • Technical Aspects: good
  • Clinical applications:

3.1. Parenchyma imaging: demonstrates more a technical feasibility, but does not describe any significance in clinical application / need.

3.2. Nodule imaging: Dito. In addition, please incorporate and discuss following article:

Jungblut L, Blüthgen C, Polacin M, Messerli M, Schmidt B, Euler A, Alkadhi H, Frauenfelder T, Martini K. First Performance Evaluation of an Artificial Intelligence-Based Computer-Aided Detection System for Pulmonary Nodule Evaluation in Dual-Source Photon-Counting Detector CT at Different Low-Dose Levels. Invest Radiol. 2021 Jul 28. doi: 10.1097/RLI.0000000000000814. Epub ahead of print. PMID: 34324462.

3.3 Lung cancer screening – 3.6 Distal Airways and bronchial imaging

Please adhere to a common citation of researchers (Zhou et al vs Rolf Symons et al.)

Otherwise, same as above

3.7 Pulmonary vascularization imaging

Rather short, there seem to be no relevant data regarding PCD-CT imaging of the pulmonary vasculature, e.g. for pulmonary embolism, CTEPH etc. The inherit advantage of PCD-imaging is in my regard more than just increase of spatial resolution for peripheral pulmonary vessels, but includes features such as VMI, Iodine map etc. Please expand and discuss

3.8 Perspectives

There are several newer publications referring to features of k-edge imaging, which are mainly conducted outside of the chest or in phantoms but could be interesting to discuss, e.g.:

Sartoretti T, Eberhard M, Nowak T, Gutjahr R, Jost G, Pietsch H, Schmidt B, Flohr T, Alkadhi H, Euler A. Photon-Counting Multienergy Computed Tomography With Spectrally Optimized Contrast Media for Plaque Removal and Stenosis Assessment. Invest Radiol. 2021 Mar 5. doi: 10.1097/RLI.0000000000000773. Epub ahead of print. PMID: 33660630.

Sartoretti T, Eberhard M, Rüschoff JH, Pietsch H, Jost G, Nowak T, Schmidt B, Flohr T, Euler A, Alkadhi H. Photon-counting CT with tungsten as contrast medium: Experimental evidence of vessel lumen and plaque visualization. Atherosclerosis. 2020 Oct;310:11-16. doi: 10.1016/j.atherosclerosis.2020.07.023. Epub 2020 Aug 5. PMID: 32861961.

Critical reflection is mostly missing in the article. Would recommend thorough discussion of technical limitations, missing evidence that the technical advantages really translate into a clinical meaningful and relevant gain of information.

Language: some strange reading sentences, e.g. “But DECT systems requires major changes in their architecture compared to single energy CT” in Chapter 2.3. Recommend linguistic revision by a native speaker to improve readability and text flow.

Reviewer 2 Report

There are several grammatical/typographical mistakes that I request the authors to carefully review and correct. For example, "Rolf Symons et al. [38]" should actually be written as "Symons et al. [38]", "idose algorithm reconstruction (idose 9)" should actually be "idose reconstruction algorithm", "fibrosis key signs" should be written as "key signs of fibrosis". I am not going to list all these grammatical errors since it is not the primary responsibility of a manuscript reviewer, I will focus on scientific content and ask the authors to correct these errors or take help from a language editor.

The authors have done a nice job in selecting a range of important articles in the literature pertaining to PCD-CT, for their review on lung PCD imaging. However, most of the figures in the manuscript are primarily from the authors own work (from a single CT vendor), and should be diversified by including figures from other articles. This will require acquiring copyright permission from other journals, which is a standard procedure for review articles. This will strengthen the manuscript further, and could serve as an important reference for the PCD imaging community. 

Introduction - second paragraph: "Semiconductor materials such as cadmium, telluride, or zinc" makes it sound like elemental composition used in the detectors, but the detectors are rather made of compounds such as cadmium-telluride or cadmium-zinc-telluride. Please rephrase. 

Authors state (in the abstract and in the manuscript) that detector pixel sizes of approximately 250 microns are possible. There are reports of pixel sizes smaller than 250 micron (e.g. in reference 16) for whole-body human imaging. Please update the text and references accordingly. 

"In addition, they presented for the first time the feasibility of bicolor K-edge imaging, i.e. simultaneous differentiation of two different contrast agents, using gold nanoparticles and an iodinated contrast agent (Figure 12)." - Please clarify here that this was an animal study, and not a patient study. Also, Figure 12 is grayscale, and the use of the term "bicolor" in the text may confuse readers. Either include a bicolor map in Figure 12, or use "dual-contrast K-edge imaging" instead of "bicolor K-edge imaging."  

3.6 Distal airways and bronchial imaging 
"Bartlett et al [29] studied on healthy volunteer (22 patients)" - couple of errors here 1. Healthy volunteers cannot be termed as "patients" and 2. Bartlett et al. did not study healthy volunteers, they have clearly mentioned in their article that the patients were scanned for clinical indications such as pulmonary nodules, malignancy, pneumonia, ILD and bronchiectasis. The authors should carefully consider such important details when reviewing other articles to avoid misinformation.  

Section 4 - Perspectives: This section currently appears to be an introduction to K-edge imaging and nanoparticles, rather than providing perspectives focused on lung imaging. In my opinion, since the focus of the article is lung imaging, the authors should provide perspectives specific to lung imaging, and their impression of how this technology could improve lung imaging and potentially lead to new applications with improved diagnostic capabilities. The current introduction/elaboration about K-edge imaging and nanoparticles, albeit a very interesting topic, is too generic and seems to be a bit out of place/context for the manuscript that maintains a good content flow about lung imaging until section 4. The authors can shorten the K-edge imaging perspective (Figure 12 can be retained due to its relevance to lung imaging), and focus more on lung diseases that can benefit from high spatial resolution, quantitative capabilities and improved CNR offered by PCD-CT.